# Non-Alcoholic Pearl Millet Beverage Innovation with Own Bioburden: *Leuconostoc mesenteroides*, *Pediococcus pentosaceus* and *Enterococcus gallinarum*

**DOI:** 10.3390/foods10071447

**Published:** 2021-06-22

**Authors:** Victoria A. Jideani, Mmaphuti A. Ratau, Vincent I. Okudoh

**Affiliations:** 1Biopolymer Research for Food Security, Department of Food Technology, Faculty of Applied Sciences, Bellville Campus, Cape Peninsula University of Technology, Bellville 7535, South Africa; ratauma@vodamail.co.za; 2Bioresource Engineering Research Group, Department of Biotechnology, Faculty of Applied Sciences, District-Six Campus, Cape Peninsula University of Technology, Bellville 7535, South Africa; okudohv@cput.ac.za

**Keywords:** pearl millet, lactic acid bacteria, *Leuconostoc mesenteroides*, *Pediococcus pentosaceus*, *Enterococcus gallinarum*, fermentation

## Abstract

The appropriate solution to the problem of quality variability and microbial stability of traditional non-alcoholic pearl millet fermented beverages (NAPMFB) is the use of starter cultures. However, potential starter cultures need to be tested in the production process. We aimed to identify and purify bioburden lactic acid bacteria from naturally fermented pearl millet slurry (PMS) and assess their effectiveness as cultures for the production of NAPMFB. Following the traditional *Kunun-zaki* process, the PMS was naturally fermented at 37 °C for 36 h. The pH, total titratable acidity (TTA), lactic acid bacteria (LAB), total viable count (TVC) and the soluble sugar were determined at 3 h interval. The presumptive LAB bacteria were characterized using a scanning electron microscope, biochemical tests and identified using the VITEK 2 Advanced Expert System for microbial identification. The changes in pH and TTA followed a non-linear exponential model with the rate of significant pH decrease of 0.071 h^−1^, and TTA was inversely proportional to the pH at the rate of 0.042 h^−1^. The Gompertz model with the mean relative deviation modulus, 0.7% for LAB and 2.01% for TVC explained the variability in microbial growth during fermentation. The LAB increased significantly from 6.97 to 7.68 log cfu/mL being dominated by *Leuconostoc, Pediococcus, Streptococcus* and *Enterococcus* with *an* optimum fermentation time of 18 h at 37 °C and 4.06 pH. *L. mesenteroides* and *P. pentosaceus* created an acidic environment while *E. gallinarum* increased the pH of the pearl millet extract (PME). Innovative NAPMFB was produced through assessment of LAB from PMS to PME fermented with *L. mesentoroides* (0.05%) and *P. pentosaceus* (0.025%) for 18 h, thereby reducing the production time from the traditional 24 h.

## 1. Introduction

Fermentation is an ancient method of food preservation and due to its nutritional value as well as a variety of sensory attributes, it is popular in many cultures [1]. Furthermore, fermentation destroys undesirable components, resulting in food safety, extension of product shelf life, protein and carbohydrate digestibility, dietary fibre modification and enhancement of vitamins and phenolic compounds [1,2]. However, the traditional fermentation process is spontaneous and uncontrolled while the products are obtained under local climatic conditions, resulting in variable sensory characteristics and quality [1]. Innovative fermentation technology of the traditional production processes could solve the problem of food safety and malnutrition in some countries where poverty, malnutrition and infant mortality are common.

Production of non-alcoholic beverages from cereals is common in many cultures. Popular in Eastern Europe and the Middle East, Bulgaria, Albania and Turkey is *Boza,* a thick, sweet-sour, low-alcoholic beverage, with high lactic acid bacteria content and probiotic properties made from fermented barley, oats, millet, maize wheat or rice [3,4,5,6]. *Bushera* is traditionally prepared in the Western Highlands of Uganda with decreased phenolic and tannin content, and the probiotic *Levilactobacillus brevis* [7]. *Kunun-zaki* is a sweet-sour and creamy drink, widely manufactured in West Africa similar to the gruels/drinks *mahewu* and *baganiya* from Sudan [3]. The beverages are consumed while the lactic acid bacteria fermentation is in progress and similar preventive and therapeutic health effects have been reported. Consumption of *Boza* improves colonic health and decreases plasma cholesterol [8]. *Bushera* and *kunun-zaki*, are believed to enhance lactation of nursing mothers and may substitute mother milk due to their high nutritive and pro/prebiotic content [3]. Although the starting raw materials for *boza*, *bushera and kunu-zaki* may differ, their production processes are similar, involving cereal soaking, germination, milling, boiling, sieving and lactic acid fermentation.

*Kunu-zaki* is a non-alcoholic beverage mainly from pearl millet. It is a low viscous, milky cream with a sweet-sour taste [8]. The production process involves steeping of millet grains, wet milling with spices (ginger, cloves and pepper), partial gelatinisation of the paste, mixing with ungelatinized paste, fermentation, sieving, mixing with sugar and bottling. The processing stage of sieving after fermentation is a critical control point and any process improvement should prevent hazards after this step. Obadina et al. [9] investigated the effect of millet grain steeping duration on the quality of *kunun-zaki* and reported that the millet grains should be steeped for 36 h. The production is often labour intensive involving rudimentary equipment resulting in a variable quality beverage with a short shelf-life. Following the production process of *kunun-zaki*, the slurry obtained after partial gelatinisation of the paste and mixed with the ungelatinised paste was used as a growth medium in this study.

Lactic acid bacteria (LAB) include the genera *Carnobacterium*, *Enterococcus*, *Levilactobacillus*, *Lactococcus*, *Trichococcus*, *Leuconostoc*, *Melissococcus*, *Oenococcus*, *Pediococcus*, *Streptococcus*, *Tetragenococcus*, *Vagococccus* and *Weissella* [10,11]. Some of these bacteria could be probiotics producing different antimicrobials such as acetic acid, carbon dioxide and bacteriocins which are of interest. It is generally agreed that LAB are responsible for the acid and flavour development of many fermented cereal foods [11]. LAB have been isolated from different traditional foods but the organisms differ from region to region and house to house as a result of differences in raw materials, leading to quality variations. The appropriate solution to the problem of quality variability and microbial stability of the product is the use of starter cultures [12] because it (1) ensures the rapid initiation of broth acidification, (2) reduces the risk of potential spoilage or pathogenic bacteria, thereby producing desirable flavours and (3) effectively control the fermentation process. However, potential starter cultures need to be tested in the production process. Our approach was a systematic study of the LAB involved in the fermentation of the pearl millet slurry to pearl millet extract for a reduction in production time. Our objective was to isolate the LAB involved in the natural fermentation of pearl millet slurry and assess their potential for the production of optimum non-alcoholic pearl millet beverage.

## 2. Materials and Methods

### 2.1. Sources of Materials and Equipment

Pearl millet was purchased from Agricol in Cape Town, South Africa. Ground ginger was supplied by Deli Spices in Cape Town, South Africa. Sprouted rice flour was obtained from the Department of Food Science and Technology, Cape Town, South Africa. The equipment were from the AgriFood Technology Station and the Department of Food Science and Technology, Cape Peninsula University of Technology.

### 2.2. Production of Pearl Millet Slurry and Fermentation

The production process for pearl millet slurry is detailed in Figure 1. The pearl millet flour (200 g) was hand mixed with 250 g water and left to hydrate for 3 h at ambient temperature (approximately 25 °C). The hydrated paste was divided into two unequal portions (¼ and ¾). The ¾ paste was gelatinised with 1000 mL boiling water and cooled to 40 °C. The ¼ paste was hand mixed with 10 g ground ginger, 30 g sprouted rice flour and 50 mL cold water. The two portions (¼ and ¾) were mixed. Aliquots (45 mL) of the slurry were distributed into sterilized 100 mL Schott bottles and left to ferment at 37 °C for 36 h in a water bath with a shaker set at 32 rpm. Samples were drawn at 3 h interval during the fermentation and analysed for pH, total titratable acidity, total soluble sugar and microbial population.

### 2.3. Physicochemical Analysis of Pearl Millet Slurry during Fermentation

The pH of the pearl millet slurry (PMS) (10 mL) was measured in triplicates using Hanna Edge glass electrode pH meter standardised with pH buffer solution of 4, 7 and 10. The total titratable acidity (TTA) was determined in triplicates by titrating 10 mL of the fermenting pearl millet slurry with 0.1 M NaOH using phenolphthalein as an indicator until a light pink colour appears. The TTA was expressed as percent lactic acid [13]. Equation (1) was used to calculate the acidity, 0.1 M NaOH equivalent to 90.08 mg lactic acid.
(1)$$\mathrm{TTA}\;\left(\%\;\mathrm{lactic}\;\mathrm{acid}\right)=\frac{\mathrm{mL}\;\mathrm{NaOH}\;\times \;N\;\mathrm{NaOH}\;\times \;M.E}{\mathrm{volume}\;\mathrm{of}\;\mathrm{sample}\;\times \;1000}\;\times \;100$$
where ml NaOH = volume of NaOH (mL), N NaOH = molarity of NaOH, M.E = the equivalent factor of lactic acid being 90.08 mg, 1000 = factor used to convert the M.E which is normally in mg to grams, and 100 used to express the lactic acid concentration in percentage.

The method of AOAC 982.14 as described by [14] was used to determine the total soluble sugars in pearl millet slurry (PMS) during fermentation.

### 2.4. Enumeration of Bacteria in Pearl Millet Slurry during Fermentation

Pearl millet slurry (PMS) (45 mL) was added into 100 mL Schott bottles and thoroughly mixed by shaking for 1 min. Dilutions of PMS were carried out by transferring 10 mL to a bottle containing 90 mL sterile ¼ strength of Ringer solution [14,15] to give 10:100 dilutions followed by a 10 fold serial dilution from 10-1 to 10-10. Each dilution was sub-cultured in triplicate. A portion of the sample dilution (1 mL) was added into 15 × 100 mm plastic Petri plates containing cooled molten agar, mixed and left to solidify. Lactic acid bacteria (LAB) were plated on deMan Rogosa and Sharpe (MRS) agar (Merck HG00C107.500) [13,16] under anaerobic condition using Anaerobic Gas-Pack system and anaerobic indicator strips at 30 °C for 48 h [13,17,18]. The total viable count (TVC) was enumerated on plate count agar (PCA) [Merck HG 0000C6.500] and incubated aerobically at 37 °C for 48 h. After incubation, Petri plates with colonies between 30 and 300 were counted. All microbiological data were expressed in the logarithm of colony-forming unit per ml (log CFU/mL).

### 2.5. Isolation and Identification of Lactic Acid Bacteria in Pearl Millet Slurry during Fermentation

Pearl millet slurry (PMS) (45 mL) was homogenized in centrifuge tubes using vortex at 5 speeds for 30 s and 1 mL was transferred aseptically into a 9 mL of ¼ strength Ringer solution and mixed thoroughly. Serial dilutions (10-1 to 10-4) were carried out and a 0.1 mL portion of the appropriate dilutions spread onto deMan Rogosa and Sharpe (MRS) agar plates. Besides, 1 mL of the serial dilution (10-1 to 10-4) was pour-plated into MRS agar. Each dilution was cultured in triplicate. The plates were incubated anaerobically for 48 h at 30 °C. Distinct colonies grown on and/or in MRS plates with 30–300 colonies were harvested and sub-cultured on to fresh MRS agar and incubated for 48 h at 30 °C. Presumptive lactic acid bacteria (LAB) colonies were further sub-cultured in triplicates on MRS agar plates and anaerobically incubated for 48 h at 37 °C.

Presumptive LAB isolates on MRS agar were examined for Gram reaction, catalase reaction, production of CO_2_ from glucose using hot loop test and gas production using 3% H_2_O_2_ [19]. Cell morphology was examined by a compound microscope and scanning electron microscope (SEM). The growth of isolates at 4, 10, 45 °C and 6.5% NaCl concentration in MRS agar were evaluated after 48 h. The colonies were identified using Vitek 2 compact system. The VITEK 2 Advanced Expert System gram-positive (GP) cards for microbial identification were used to identify the isolates (*Enterococcus*, *Lactococcus*, *Leuconostoc*, *Pediococcus*, *Streptococcus* and *Vagococcus*) to species while anaerobic cards were used to identify *Lactobacillus* to species. Vitek 2 compact system uses the principle of flurogenic method for microbial identification using a 64 well cards. The test cards used for microbial identification are divided into Gram-negative (GN) cards, Gram-positive (GP) cards, anaerobic (ANC) cards, *Neisseria* and *Haemophilus* (NH) and yeast (Yst). An inoculum from isolated pure cultures were homogenised in a 3 mL of 0.45% NaCl saline of pH 4.5–7.0 using a sterile swab to the density equivalent of 0.5–0.63 McFarland standard. The turbidity was verified using Vitek 2 DensiCheck Plus equipment calibrated with 0, 0.5, 2 and 3 McF standards. The homogenised specimens in test tubes together with the selected GP/ANC cards were placed into cassette, scanned and loaded into the Vitek 2 compact system and run following the manufacturer operating procedure. The cards were filled with the homogenised specimens by vacuum created within the equipment, sealed and placed into the machine incubator (35 °C) [20,21]. The cards were exposed to a kinetic fluorescence measurement every 15 min for 2–8 h and the results read against GP and ANC database in the equipment, and the results were made available automatically while the cards ejected into waste container.

The isolates were grown in 500 mL deMan Rogosa and Sharpe broth at 30 °C for 60 h. The broths were hand-mixed thoroughly and 2 mL of the broth mixed with 1 mL of 10% skim milk [22] in 5 mL bench-top freeze-dryer vials. The samples were frozen in an ultra-freezer (Glacier, −86 °C ultralow temperature freezer) at −76 °C for 12 h then freeze-dried using BenchTop-Pro with Omnitronics (VirTis SP Scientific) freeze dryer. The dried samples were sealed under vacuum and stored in the freezer at −18 °C.

### 2.6. Lactic Acid Bacteria Preparation for Scanning Electron Microscope Imaging

The scanning electron microscope images were used to verify the identified lactic acid bacteria based on morphology. The methods of [23,24] were used to obtain images of lactic acid bacteria (LAB) using a scanning electron microscope (SEM). LAB colonies were grown in MRS broths at 30 °C for 36 h. The broth was mixed thoroughly for 1 min and few drops (4–5) placed on 0.45 µm filters and then left to air dry at room temperature for 30 min. The specimens were then fixed using 2.5% glutaraldehyde in phosphate-buffered saline (PBS) with a pH of 7.2 for 30 min at 4 °C. The specimens were fixed using osmium tetroxide (OsO_4_) for 1 to 2 h before dehydration in a series of ascending different ethanol concentration (30, 50, 70, 80 and 100%) for 15 min at each concentration. The final stage in 100% ethanol was repeated twice. The specimens were then critically-point dried at 1072 psi and 31 °C, then coated with gold before viewing under Zeiss MERLIN FE-SEM. Beam conditions during imaging were 5 kV accelerating voltage, 250 pA probe current, with a working distance of approximately 4 mm.

### 2.7. Experimental Design for the Effect of Bioburden Lactic Acid Bacteria on Pearl Millet Extract

The three isolated lactic acid bacteria (*L. mesenteroides*, *P. pentosaceus* and *E. gallinarum*) from the pearl millet slurry were assessed for their effect on pearl millet extract (PME), a modification of the pearl millet slurry extraction method to reduce production time. Pearl millet extract (produced by hydrating pearl millet flour with water (1:10)), with 15% sprouted rice flour, 10% ground ginger and 0.6% pectin were pasteurized, cooled to 40 °C and inoculated with *L. mesenteroides*, *P. pentosaceus* and *E. gallinarum*. The inoculation followed a randomized three-level augmented factorial design (19 runs) each culture at two levels (0.05, 0.1%) with three center points to determine the optimum culture. Each design run experiment was conducted in triplicate. The inoculum was fermented for 18 h at 37 °C. The generalized linear model (Equation (2)) was used to determine the effect of the purified cultures of lactic acid bacteria (LAB) on the pH, total titratable acidity (TTA) and viscosity of the beverage. The model obtained was used to simulate 1000 cases using Monte Carlo simulation to establish the influence of the LAB on the pH, TTA and viscosity.
(2)$${Y=ß}_{0}{+ß}_{1}X_{1}{+ß}_{2}X_{2}{+ß}_{3}X_{3\;}{+ß}_{12}X_{1}X_{2}{+ß}_{13}X_{1}X_{3}{+ß}_{23}X_{2}X_{3}$$
where Y represents the estimated parameter response (pH, lactic acid or viscosity). ß_0_ represents the overall mean (intercept), β_1_, β_2_ and β_3_ are the main effects for *L. mesenteroides*, *P. pentosaceus* and *E. gallinarum*, respectively. β_12_, β_13_ and β_23_ are the interactive effect of the lactic acid bacteria. X1, X2 and X3 represent independent factors, *L. mesenteroides*, *P. pentosaceus* and *E. gallinarum*, respectively.

### 2.8. Effect of L. mesenteroides and P. pentosaceus on the pH, Total Titratable Acidity (TTA) and Viscosity of the Pearl Millet Extract

A two-level factorial design for *L. mesenteroides* and *P. pentosaceus* each at two levels (0.05 and 0.10%) augmented with a centre point was used to evaluate their effects on the acceptability (benchtop sensory), pH, total titratable acidity and viscosity of the PME. The experimental design was run randomly in triplicate. A benchtop sensory revealed that the taste of the beverage was not acceptable. Thereafter, *L. mesenteroides* and *P. pentosaceus* were used in combination at 0.05% each and combination at 0.05 and 0.025%, respectively. A benchtop sensory was used to evaluate the taste of the beverage.

### 2.9. Production of Optimal Non-Alcoholic Pearl Millet Beverage

Pearl millet extract (PME) (pearl millet flour with water (1:10)), (1000 mL) was weighed into a 5 L plastic beaker and blended with 0.6% pectin, 0.1% sodium citrate, 1% sunflower lecithin and 5% white sugar at 6600 rpm for 7 min using a Silverson L4RT homogenizer while slowly adding the dry ingredients. The mixture was pasteurised in a pot at 85 °C for 15 min and hot-filled into 100 mL Schott bottles. The bottles were rapidly cooled to 25 °C in ice blocks and tap water. The extract was then aseptically inoculated with a mixture of *L. mesenteroides* (0.05%) and *P. pentosaceus* (0.025%); fermented for 18 h at 37 °C. The resulting non-alcoholic pear millet beverage (NAPMB) was then chilled at 4 °C until required.

### 2.10. Determination of the Viscosity of Non-Alcoholic Pearl Millet Beverage

The change in viscosity of pearl millet beverage over time was determined using Rheolab QC (Anton Paar) with temperature device C-PTD 180/AIR/QC and measuring system CC27. The beverage (18 mL) was poured into an upward projected sample cup and analyzed following the manufacturer’s instruction at 5 °C and 22 °C for 5 min. In all runs, the shear stress (τ) was set at 20 Pascal. The average of the triplicates was used.

### 2.11. Data Analysis

The results were reported as mean ± standard deviation of three triplicate runs. Multivariate Analysis Of Variance (MANOVA) was used to determine the mean difference between treatments at *p* = 0.05. Duncan’s multiple range test was used to separate means where differences exist using version 23 of IBM SPSS (IBM, 2015). The statistical relationships between the dependent variables pH, total titratable acidity, total viable count, yeast and mould during the fermentation of pearl millet slurry for the production of non-alcoholic pearl millet beverage were determined using Pearson correlation

## 3. Results and Discussion

### 3.1. Effect of Fermentation Time on the pH and Total Titratable Acidity (TTA) of Pearl Millet Slurry

The physicochemical and microbial characteristics of the pearl millet slurry during fermentation is outlined in Table 1. The changes in pH and TTA followed a non-linear exponential model as in Equation (3).
(3)$$pH=a-bexp^{-ct}$$
where *a* = horizontal asymptote; *b* = *a* − y_intercept_ or difference between the horizontal asymptote and the value of y when *x* = 0; *c* = the rate of constant (h^−1^) and *t* = fermentation time (h).

The models for pH and TTA accounted for 97.1% and 98.1%, respectively, of the variability in the pH and TTA, respectively. The changes in pH and total titratable acidity of pearl millet slurry (PMS) over the 36-h fermentation are shown in Figure 2. There was a significant (*p* ˂ 0.05) decrease in pH during the fermentation ranging from 6.37 to 3.77 in 36 h due to the increase in the population of lactic acid bacteria (LAB), which fermented glucose to lactic acid and carbon dioxide. The pH kinetics of the fermented millet slurry is indicated in Table 2. The rate of pH decrease during fermentation was 0.071 h^−1^ with a lower asymptote of 3.38. At the beginning of fermentation, the LAB were in the lag phase (0–3 h), thereafter the organisms exponentially produced significant acid until 21 h followed by a stationary phase (24–30 h). The decrease in pH could be due to the build-up of hydrogen ions as microorganisms break down starch. Meanwhile, the stationary phase could have been caused by the exhaustion of nutrient and the build-up of waste by LAB.

These results were in agreement with the report [9] that a decrease in pH during the fermentation of *Kunun-zaki* was caused by the formation of organic acid from carbohydrates and other food nutrients.

The total titratable acidity (TTA) [expressed as % lactic acid] increased from 0.12% at the start of fermentation to 0.53% at the end of 36 h. The TTA kinetics of the fermented millet slurry is indicated in Table 1. The rate of TTA increase during fermentation was 0.042 h^−1^ with a horizontal asymptote of 0.663%. There was a significant (*p* ˂ 0.05) change in TTA over the 36 h fermentation time. This could be attributed to the decrease in pH as the concentration of acid increased. The increase in LAB produced more lactic acid from the fermentation of sugars. The increase in acidity could be the cause of a sweet-sour taste of non-alcoholic pearl millet beverage in agreement with [25] during the fermentation of *Masvusvu* and *Mangisi*. Also, after 18 h of fermentation, there was no significant change in the pH and TTA. This is in agreement with the LAB growth curve which peaked after 18 h. Thus, the optimum fermentation time for the slurry could be 18 h at 37 °C with the pH expected to be 4.06.

### 3.2. Soluble Sugar Kinetics of Pearl Millet Slurry during Fermentation

The main soluble sugar identified in pearl millet slurry (PMS) was glucose which ranged from 0.54 to 2.05%. Figure 3 shows a significant (*p* ˂ 0.05) increase in glucose content up to 27 h, after which there was a quadratic drop due to the breakdown of the pearl millet starch. The glucose kinetics (R^2^ = 0.947) could be expressed as Glucose (%) = 0.584 + 0.107t − 0.002t^2^, where t = fermentation time in h.

The quadratic model accounted for 94.7% of the variability in glucose. The increase in glucose during PMS fermentation could be attributed to the decrease in starch caused by the action of α- and β-amylase activities. During fermentation, enzymes hydrolyse starch to produce monomeric sugar glucose. Although there was an increase in glucose content from the onset of fermentation, the glucose did not significantly increase after 20 h. This could be due to the acidification (low pH) of the slurry which terminates the activity of alpha-amylase by the build-up of tannins [26]. Tannins are natural polyphenols found in most cereal grains. They can act as antioxidants together with phytic acid and phenols [27]. Similarly [26] reported glucose as the main soluble sugar which gradually increased in the first 20 h during the fermentation of pearl millet flour for the production of *Lohoh* bread [28] also identified 0.5% glucose during the fermentation of *Kunun-zaki*. Therefore, fermentation time affected the glucose content of pearl millet slurry.

### 3.3. Kinetics of Lactic Acid Bacteria and Total Viable Microbes in Pearl Millet Slurry during Fermentation

The lactic acid bacteria (LAB) and total viable count (TVC) were modelled using the Gompertz equation (Equation (4) as modified by [29].
(4)$$Log\;\mathrm{CFU}/\mathrm{mL}=K+A\cdot \exp\left\{-\exp\left\{\left[\left(\mu_{max}\cdot 2.7182\right)\cdot \frac{\lambda -t}{A}\right]+1\right\}\right\}$$
where *K* = initial level of bacterial count (log CFU/mL), *A* = increase in log CFU/mL between time = 0 and the maximum population density at the stationary phase, µ_max_ = maximum growth rate (Δlog (CFU/mL)/h, λ = lag time (h) and *t* = fermentation time (h).

The model parameters are detailed in Table 3. The goodness-of-fit was evaluated by the mean relative deviation modulus, 0.7% for LAB and 2.01% for TVC. The model explained the variability in microbial growth and could be used to explain the trend in growth during fermentation. The growth pattern of pearl millet slurry during fermentation is shown in Figure 4.

There was an apparent lag time of 3.9 h, after which the growth of LAB significantly (*p* < 0.05) increased until 18 h. This trend is in agreement with those reported by [30] during the preparation of *Chibwantu*. This similar concept was explained by [31]. Although most LAB tolerates low pH, certain strains may have been retarded [32]. The growth of *Leuconostoc* and lactic streptococci rapidly drops the pH during fermentation to 4.0–4.5 and then retard their growth, thus giving way to subsequent bacteria. Lactobacilli spp. and pediococci spp. succeeded leuconostoc bacteria during fermentation, resulting in their growth retardation when the pH reached 3.5. These results are similar to those reported for spontaneous fermentation of millet by [16].

The lag phase was followed by the exponential increase of LAB from 6.76 (9 h) to 7.87 log CFU/mL (12 h) in 3 h and then accelerated to the highest count of 8.10 log CFU/mL after 15 h. During this growth phase, the cells surviving the acidic environment could be growing and dividing at the maximum rate. There was a slight decrease in the LAB to 7.79 log CFU/mL (18 h), then the organisms remained stationary for 9 h (18–27 h) with an average of 7.95 CFU/mL. Since this is a batch fermentation the growth of organisms could have been limited by depletion of nutrients, build-up of inhibitory metabolites or end-product (lactic acid) and/or shortage of biological space. Thereafter, there was a death phase as the cells started to decrease from 7.97 (30 h) to 7.68 log CFU/mL (36 h). A similar trend of LAB growth was reported by [23] during the fermentation of *Umqombothi*.

There was a significant (*p* ˂ 0.05) growth in total viable count (TVC) over a 36 h fermentation period (Figure 4b). The TVC cells accelerated from 6.98 log CFU/mL at the onset to 7.38 log CFU/mL in 3 h. This was followed by a significant (*p* < 0.05) exponential increase to 7.92 log CFU/mL (6 h). The lag phase was not visible during the growth of TVC. This may be caused by the rapid growth of mixed microbes which dominated the spontaneous fermentation of the pearl millet slurry. At this stage, certain bacteria other than LAB could be growing at a faster rate. This could also have been caused by mixed microbes not taking long to adapt to the new environment. The growth went into a stationary phase which lasted for 6 h (6–12 h). The numbers of cells during the death phase were reduced from 7.88 to 7.51 0.04 log CFU/mL (12–15 h).

The decrease in cells may be due to the build-up of lactic acid caused by mostly LAB. There was a significant (*p* < 0.05) reduction in cells (death phase) after 27 h for 3 h followed by an acceleration phase for 3 h (30 h). Bacteria not tolerating low pH could have caused a decrease in TVC. The shift-up and shift-down could also be caused by the environmental conditions resulting in competition for survival among different species of LAB. In particular, the extended stationary phase could have led to cell reduction (death) with no new nutrients fed into the system. These results are similar to those reported by [25] during the fermentation of *Masvusvu* and *Mangisi.*

There was a very strong, negative linear relationship between TTA and pH of the beverage during fermentation (*r* = −0.975, *p* < 0.05). Meanwhile, the TTA had a moderate positive relationship between lactic acid bacteria (LAB) count (*r* = 0.440, *p* < 0.05). The pH had a negative moderate relationship with the LAB count (*r* = −0.535, *p* < 0.05). The LAB count had a positive weak and very weak relationship with the TVC and YM, respectively. Meanwhile, the TVC had a positive weak relationship with the YM. A moderate-strong linear relationship was between the TTA and pH, LAB and pH. These results further indicated that during succession fermentation of glucose by LAB to lactic acid, the pH dropped due to the built-up of hydrogen ion. The decrease in pH thus increased TTA.

### 3.4. Lactic Acid Bacteria Associated with Pearl Millet Slurry Fermentation

The isolates identified from pearl millet slurry (PMS) during fermentation over 36 h are shown in Table 4. Lactic acid bacteria (LAB) from the genera *Leuconostoc, Pediococcus* and *Enterococcus* were the main species involved in the fermentation. The *Leuconostoc mesenteroides* ssp. *dextranicum* (Figure 5a) is characterized by a lenticular coccoid cells in chains and *Leuconostoc pseudomesenteroides* were identified at the beginning of fermentation between the pH of 5.59 (0 h) and 6.37 (6 h). *Leuconostoc*’s presence at the beginning of fermentation may be attributed to their growth condition at pH 6.0–6.5. This is identical to the study by [33] who identified *L. pseudomesenteroides* from *Oshashikwa*, traditionally fermented milk in Namibia. The organisms were responsible for the initiation of lactic acid fermentation. These heterolactic organisms produce carbon dioxide and organic acids which rapidly lower the pH of the beverage to 4.0 or 4.5 and inhibit the development of undesirable microorganisms. The carbon dioxide produced replaces the oxygen, making the environment anaerobic [34] and suitable for the growth of subsequent organisms such as *Lactobacillus*. Besides, the anaerobic environment created by the CO_2_ has a preservative effect on the beverage since it inhibits the growth of unwanted bacteria contaminants [35].

As reported by [36], *L. pseudomesenteroides* is widely present in many fermented foods such as dairy, wine and beans while *L. mesenteroides* is associated with sauerkraut and pickled fermented products [34]. The organism produces dextrans and aromatic compounds (diacetyl, acetaldehyde, and acetoin) which could contribute to the taste and aromatic profile. These organisms were isolated by [37] from fermented Greek table olive. *Pediococcus pentosaceus* were tetra-cocci and smooth as shown in Figure 5b isolated at 0, 9, 18 and 36 h of fermentation, similar to the report by [14,38]. Although it grows between pH 4.5–8, the optimum growth is between pH 5.0 and 6.5. MRS agar was developed for LAB growth but selective for Lactobacilli especially *Leuconostoc* spp. which may or may not grow. *Pediococcus* spp. similar to *Leuconostoc* spp. and *Streptococcus* spp. growth is enhanced considerably in a microaerobic environment with 5% CO_2_. Anaerobic Gas-Pack system produces between 4 to 10% CO_2_ of which if it was above 5% could have slowed or inhibited the growth of *Pediococcus*. Microaerobic organisms also require oxygen content typically between 2–10%, whereas the Anaerobic Gas-Pack system usually creates an oxygen content of <1%. The low level of oxygen could have affected their growth or reduced their number at certain times. The genus *Pediococcus* belongs to the family *Lactobacillaceae* in the order *Lactobacillales* growing at optimum pH of 4.5–8.0. They can produce bacteriocins (antimicrobial agent), which are used as a food preservative. The bacteriocins produced inhibit the growth of Gram-positive bacteria since they attack the cytoplasmic membranes of the cell which is protected by the polysaccharide protective layer in Gram-negative cells [35]. *Streptococcus thoraltensis* cells were cocci and and also present after 6 h of fermentation (Figure 5d). The presence of *S. thoraltensis* could be through contamination of pearl millet grains and/or utensils. The organism was isolated from animal intestinal tracts of swine [39]. Several enterococci (Table 2) were isolated throughout the fermentation at different times (3–30 h) between pH 3.81 and 6.09. They became active between 12 and 30 h and are known to grow well between pH 4 and 9.6 [40]. All enterococci identified were ovoid and appeared in pairs or long chains (Figure 5). The organisms are responsible for the development of flavours due to their glycolytic, proteolytic and lipolytic activities. They have probiotic activities and have the potential as bio-preservatives. In general, enterococci are ubiquitous and are found in the environment and gastrointestinal tract of healthy animals and humans [40]. These organisms are used as starter cultures in the fermentation of food since they create unique sensory properties [40] and contribute to texture and safety [41]. *Enterococcus casseliflavus* shown in Figure 5f and *Enterococcus gallinarum* in Figure 5c were identified after 3 and 15 h. *E. casseliflavus* have been isolated from olive brines and traditional fermented food and used as starter culture [42,43]. *E. gallinarum* was isolated between 12 and 30 h at pH 3.81 to 4.68, similar to [43] who identified the organisms in Nigerian traditional fermented foods. They have lipolysis, proteolysis, bile-tolerating and low pH tolerating properties. They have hydrophobic properties and produce bacteriocins that will inhibit food pathogens and spoilage microorganisms [43]. *E. faecium* (Figure 5g) was detected at 12, 15, 18 and 21 h, while *E. faecalis* was detected after 30 h. *E. faecium* and *E. faecalis* are reported to be probiotics but their source may be through contamination.

The author [33] also reported the isolation of *E. faecium* from traditionally fermented milk *Omashikwa*. However, as reported by [40], they are suspected to be pathogenic to humans and are resistant to antibiotics.

The biochemical properties of presumptive lactic acid bacteria (LAB) isolates are shown in Table 5. All the isolates were Gram-positive, catalase-negative and did not produce gas from glucose. All cells were cocci and cocci-oval in morphology and showed no growth at 4 °C. At 45 °C there was no growth of the isolates except *E. faecium.* The inability of all the LAB isolates to grow at 4 °C could demonstrate increased glycolytic activity which could lead to the increased production of lactic acid [44]. However, the same report by [44] stated that the growth of *Lactococcus* at low temperature resulted in reduced production of lactic acid due to the reduced glycolytic activity. The inability to grow at high temperature could mean that the LAB strain has a high growth rate and lactic acid production. Their inability to grow at 45 °C are in disagreement with the report in [44]. The differences could be due to the period of incubation which was 2–4 h, whereas this study was incubated for 48 h. The isolates grew at 10 °C and 6.5% NaCl concentration except for *E. avium.* The growth of all LAB isolates except *E. avium* in 6.5% salt concentration indicated that the LAB strain could be used as a commercial starter culture. During the commercial production of lactic acid by LAB strain, alkali could be added to increase the pH and reduce an excess decrease in pH [44]. However, the same report mentioned that LAB strains grown in the presence of salt could lead to the loss of turgor pressure, leading to an effect on the physiology, enzyme activity, water activity and metabolism of the cell. These physiological properties could be used to confirm the ability of the LAB isolates to be used as starter cultures. The LAB of importance during fermentation of pearl millet slurry was from the genera *Leuconostoc, Pediococcus* and *Enterococcus.* Thus, further studies involved the utilisation of *L. mesenteroides*, *P. pentosaceus* and *E. gallinarum* were chosen as starter cultures to ferment pearl millet extract.

### 3.5. Effect of Isolated Bioburden Lactic Acid Bacteria on the pH, Total Titratable Acidity and Viscosity of Pearl Millet Extract

The pH, titratable acidity and viscosity of fermented pearl millet extract (PME) as affected by the isolated bioburden lactic acid bacteria is detailed in Table 6. The generalized linear model (GLM) for the effect of *L. mesenteroides*, *P. pentosaceus* and *E. gallinarum* is shown in Table 7. *L. mesenteroides*, *P. pentosaceus, E. gallinarum,* the interaction between *L. mesenteroides* and *P. pentosaceus* and that between *L. mesenteroides* and *E. gallinarum* had a significant effect (*p* ≤ 0.05) on the pH except for the interaction between *P. pentosaceus and E. gallinarum*. There was a significant (*p* < 0.05) increase in the pH by *L. mesenteroides*, *P. pentosaceus* and the interaction between *L. mesenteroides* and *E. gallinarum*. *E. gallinarum* and interaction effects of *L. mesenteroides* and *P. pentosaceus* had a significant (*p* < 0.05) decrease on the pH. The interaction effects of *P. pentosaceus* and *E. gallinarum* caused a non-significant decrease in the pH of the beverage.

Monte Carlo simulation of 1000 cases with input uniform distribution using the GLM indicated that *E. gallinarum* consistently increased the pH of the extract. The pH of the 95% cases were below 3.66. The sensitivity analysis indicates the degree to which the pH is sensitive to the lactic acid bacteria. The correlation tornado chart shows that pH is most strongly positively correlated with *E. gallinarum*. Overall, all the pure cultures had a significant (*p* < 0.05) effect on the pH of the pearl millet extract with *P. pentosaceus* having the highest contribution (973.9%), followed by *E. gallinarum* (655.5%) and lastly *L. mesenteroides* (132.7%). Lactic acid bacteria (LAB) in general can tolerate a wide range of pH in the presence of organic acid such as lactic acid. *L. mesenteroides* grows early during food fermentation and then superseded by the growth of other LAB. During LAB fermentation, carbohydrates are broken into lactic acid which allow for the growth of acidophilic bacteria such as *P. pentosaceus* and *E. gallinarum* [45]. *L. mesenteroides* and *P. pentosaceus* were responsible for the creation of an acidic environment while *E. gallinarum* increased the pH of the beverage. The increase in the pH could be caused by the autolysis of *E. gallinarum* as a result of the unfavorable acidic (pH 3.32 to 3.90) growth environment.

*L. mesenteroides* (heterolactic bacteria) produced the least acid unlike the homolactic bacteria *P. pentosaceus.* The heterolactic bacteria produce about 50% lactic acid, 25% acetic acid and ethyl alcohol and 25% CO_2_. In contrast, homolactic produces mainly lactic acid [46]. The CO_2_ produced replaces oxygen present in the beverage and create an anaerobic environment which gave growth to subsequent anaerobic bacteria [46]. This is in agreement with [47] who reported the growth of *Pediococcus* spp. dominated the latter stages of fermentation of maize. *P. pentosaceus* was responsible for the rapid acidification of dough. The addition of amylase-rich sprouted rice flour was necessary since chance LAB fermentation requires the enzymes to saccharify the grain starch [47]. The pH of the PME is expected to decrease as more lactic acid accumulates during fermentation, but *E. gallinarum* increased the pH based on the Monte Carlo simulation. Thus *E. gallinarum* is not a promising culture for fermenting PME.

The generalized linear model for the main effect of *L. mesenteroides*, *P. pentosaceus* and *E. gallinarum* and their interactions on the total titratable acidity (TTA) of pearl millet extract are shown in Table 8. All the cultures had a significant influence (*p* ≤ 0.05) on the TTA of the pearl millet extract. The interaction between *L. mesenteroides* and *P. pentosaceus* caused a significant (*p* < 0.05) increase in the TTA.

The TTA of the 95% cases was below 0.60%. The correlation tornado chart shows that TTA is most strongly negatively correlated with *L. mesenteroides*. *P. pentosaceus* had a high contribution (526.3%) of influence on TTA followed by *E. gallinarum* (137.3%), then *L. mesenteroides* (72.54%). The TTA was measured as the total lactic acid produced from the fermentation of starch and sugars by LAB. During fermentation, homolactic bacteria *P. pentosaceus* and *E. gallinarum* produced mainly lactic acid whereas *L. mesenteroides* produced lactic acid, CO_2_ and acetic acid/ethyl alcohol. Thus, *P. pentosaceus* and *E. gallinarum* contributed highly to the production of lactic acid. This was in agreement with [48] who reported an increase in TTA during the fermentation of non-alcoholic beverages from cereals. However, based on the generalised linear model and Monte Carlo simulation, *E. gallinarum* caused a significant increase and decrease in the pH and TTA, respectively. This is not desired for a beverage fermentation, hence the culture was eliminated.

### 3.6. Effect of Different Purified Lactic Acid Bacteria on the Viscosity of Pearl Millet Extract

Table 9 shows the generalized linear model for the main effect of *L. mesenteroides*, *P. pentosaceus* and *E. gallinarum* on the viscosity of PME. *L. mesenteroides*, *P. pentosaceus*, *E. gallinarum*, the interaction between *L. mesenteroides* and *P. pentosaceus*, the interaction between *L. mesenteroides* and *E. gallinarum*, the interaction between *P. pentosaceus* and *E. gallinarum* and the interaction between *L. mesenteroides*, *P. pentosaceus* and *E. gallinarum* had a significant influence (*p* ≤ 0.05) on the viscosity of the beverage.

The interaction between *L. mesenteroides* and *P. pentosaceus*, the interaction between *L. mesenteroides* and *E. gallinarum*, the interaction between *P. pentosaceus* and *E. gallinarum* and the interaction between *L. mesenteroides*, *P. pentosaceus* and *E. gallinarum* significantly (p < 0.05) increased the viscosity of the PME. Meanwhile, the decrease in the viscosity was caused by *L. mesenteroides*, *P. pentosaceus* and *E. gallinarum*. The interaction between *L. mesenteroides*, *P. pentosaceus* and *E. gallinarum* caused a thicker beverage than the effect of all other lactic acid bacteria (LAB). This similar concept was seen in the beverage with *L. mesenteroides* whereas *P. pentosaceus* caused the increase in viscosity. Monte Carlo simulation indicated that 95% of the cases have viscosity less than 7.80 mPa.s. The correlation tornado chart shows that viscosity is most strongly positively correlated with *P. pentosaceus*. *P. pentosaceus* had the highest contribution (74.01%) on the viscosity followed by *E. gallinarum* (50.41%) and *L. mesenteroides* (45.98%).

During cereal fermentation LAB break down starch into simpler sugars resulting in a decrease in viscosity. The viscosity of the beverage is affected by factors such as the pH, type of microorganisms and if the type of microorganisms involved in fermentation has amylase enzymes to hydrolyze starch into dextrins and sugars. In this study, high amylase sprouted rice flour (SRF) enhanced the decrease in the viscosity of the beverage. Thus, the breakdown of starch by SRF and LAB had several desirable effects on the viscosity and nutritional quality. These results are in agreement with [49] who reported a decrease in viscosity after fermentation of a traditional fermented beverage (*Boza*) at 20 °C. *L. mesenteroides*, *P. pentosaceus* and *E. gallinarum* and the interaction between *P. pentosaceus* and *E. gallinarum* caused a decrease in the viscosity of the beverage which is desired for a beverage. However, *E. gallinarum* could not be used since it causes an increase in the pH and a decrease on the TTA of the beverage. Therefore, going forward *L. mesenteroides* and *P. pentosaceus* were selected in the production of the beverage.

### 3.7. Non-Alcoholic Pearl Millet Beverage (NAPMB) Produced Using Pure Cultures of Lactic Acid Bacteria (LAB)

When cultures (*L. mesenteroides* and *P. pentosaceus*) were used individually to ferment PME at 0.05% each and in combination at 0.05%, *L. mesenteroides* alone produced a beverage with better taste compared to *P. pentosaceus.* The beverage with *L. mesenteroides* (0.05%) and *P. pentosaceus* (0.025%) produced an acceptable beverage. [50] reported that *Pediococcus* spp. are responsible for the production of diacetyl which results in ‘buttery’ aroma hence the reduction of *P. pentosaceus* to 0.025% produced acceptable beverage. The pearl millet extract [1:10 (flour:water)] mixed with pectin (0.6%), sunflower lecithin (0.1%), sodium citrate (0.1%) and fermented with *L. mesenteroides* (0.05%) and *P. pentosaceus* (0.025%) for 18 h produced a stable non-alcoholic pearl millet beverage. The next article will report on the physicochemical and nutritional quality of the beverage. These plant-originated *Lactobacillus* and *Pediococcus* strains have been reported to display in vitro probiotic effects including acid and bile tolerance, high levels of antioxidant activity, and strong adhesion to HT-29 cell [51].

## 4. Conclusions

The natural fermentation of pearl millet slurry was dominated by lactic acid bacteria (LAB) and contaminants, and their survival were in succession due to the increase in lactic acid. *L. pseudomesenteroides, L. mesenteroides ssp. dextranicum, E. gallinarum* and *P. penotosaceus* were the main fermenting LAB. Optimal non-alcoholic pearl millet beverage could be produced by fermenting the slurry for 18 h at 37 °C with expected pH of 4.06. Lactic acid bacteria (LAB) are associated with total titratable acidity (TTA) which could be used as an indicator for the survival of LAB. The pearl millet extract (1:10 (flour:water)) mixed with pectin (0.6%), sunflower lecithin (0.1%), sodium citrate (0.1%) and fermented with *L. mesenteroides* (0.05%) and *P. pentosaceus* (0.025%) for 18 h produced a stable non-alcoholic pearl millet beverage. The study has shown that a similar beverage to a traditionally prepared beverage can be produced under controlled conditions with controlled quality. The identified LAB could be developed as starter cultures for industrial production. The beverage could be industrialised and made available to urban and semi-urban dwellers. More work could be done to confirm these strains at a molecular level and to improve the taste to their desired profile.

## Figures and Tables

**Figure 1 foods-10-01447-f001:**
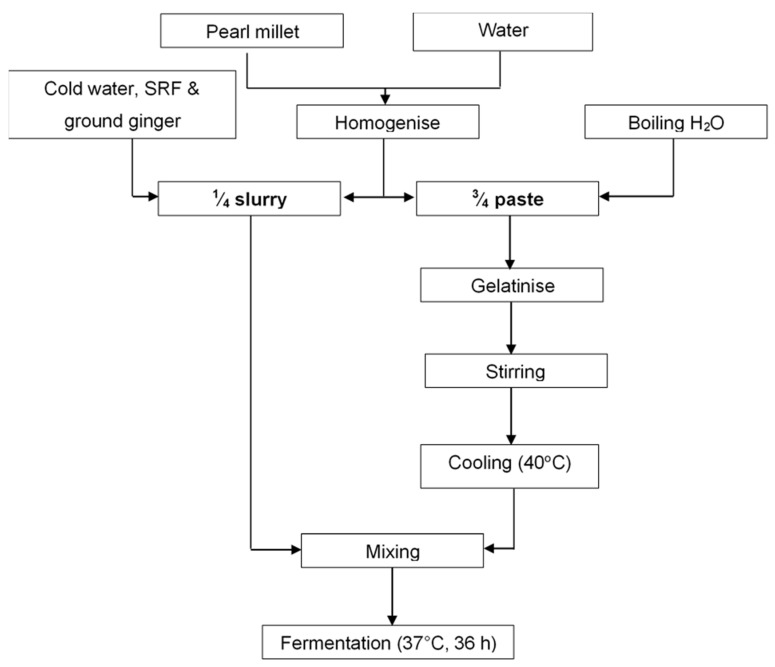
Production process for the pearl millet slurry. SRF—sprouted rice flour.

**Figure 2 foods-10-01447-f002:**
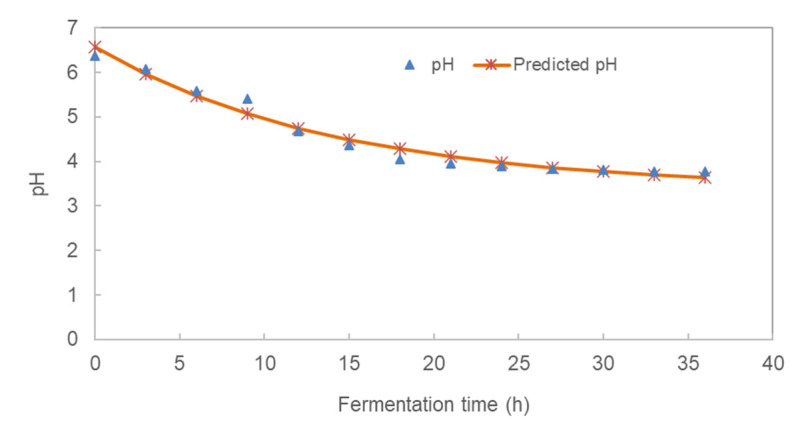
Kinetics of pH and titratable acidity of pearl millet slurry during fermentation.

**Figure 3 foods-10-01447-f003:**
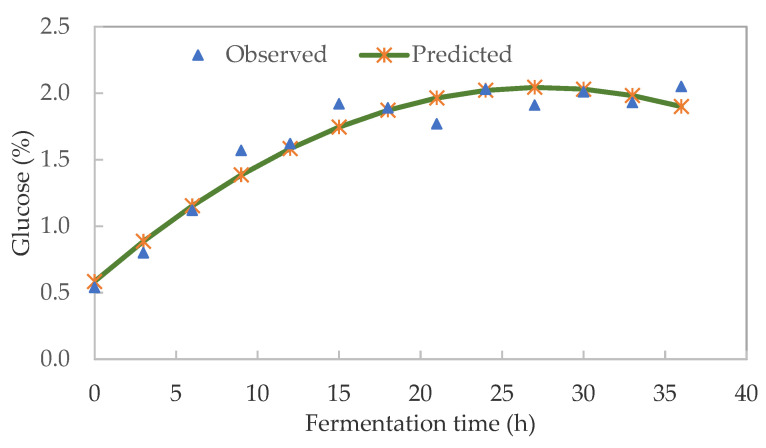
Effect of fermentation time on the glucose content of pearl millet slurry during fermentation.

**Figure 4 foods-10-01447-f004:**
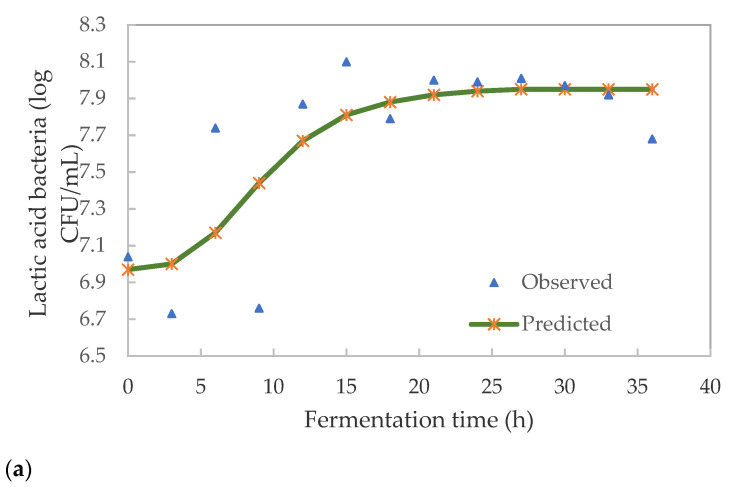
Bacterial growth curve for the fermentation of pearl millet slurry (**a**) LAB—lactic acid bacteria; (**b**) TVC—total viable count.

**Figure 5 foods-10-01447-f005:**
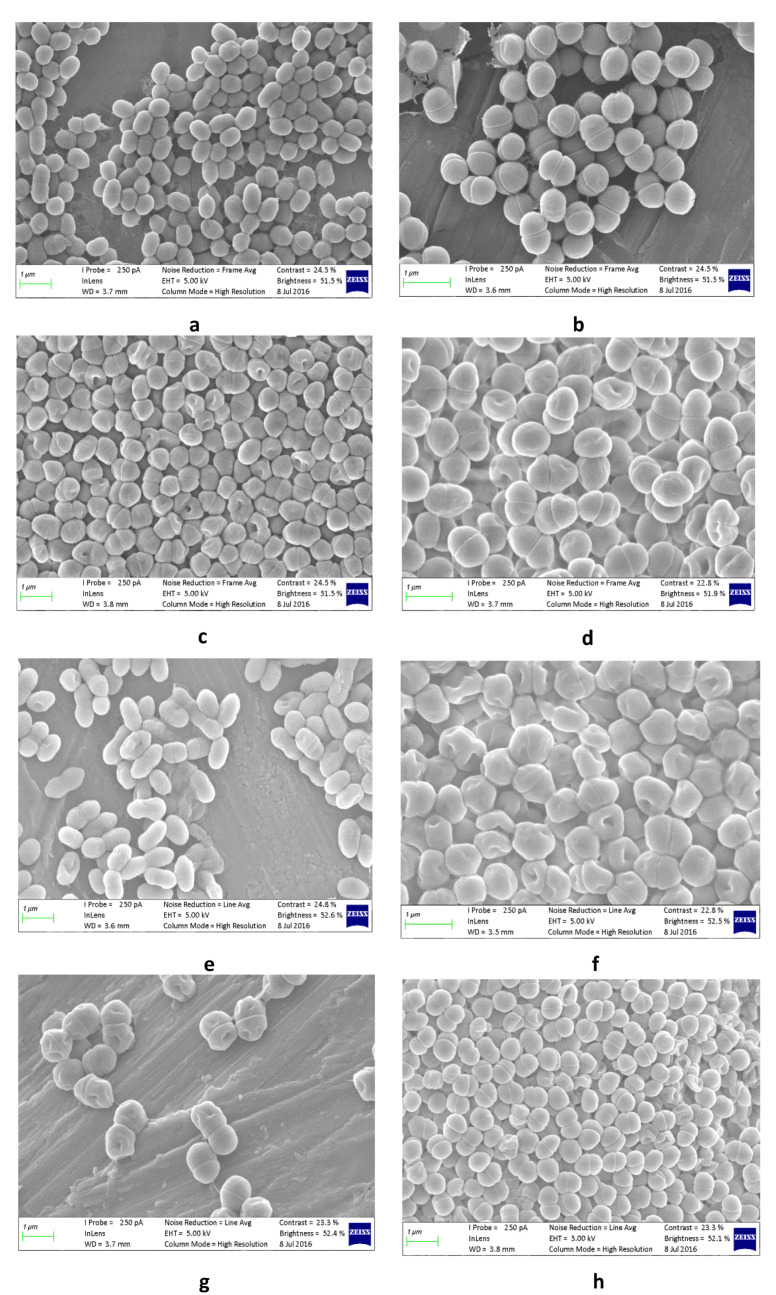
Scanning electron microscopy of bacterial cells isolated from pearl millet slurry during fermentation. Tentative identities: (**a**) *Leuconostoc mesenteroides* ssp. *dextranicum* (1 µm)*,* (**b**) *Pediococcus pentosaceus* (1 µm)*,* (**c**) *Enterococcus durans* (1 µm)*,* (**d**) *Streptococcus thoraltensis* (1 µm)*,* (**e**) *Enterococcus gallinarum* (2 µm)*,* (**f**) *Enterococcus casseliflavus* (1 µm)*,* (**g**) *Enterococcus faecium and* (**h**) *Enterococcus avium* (1 µm).

**Table 1 foods-10-01447-t001:** Physicochemical and microbial properties of pearl millet slurry during fermentation *.

Fermentation Time	pH	TTA (%)	Glucose (%)	LAB (log CFU/mL)	TVC (log CFU/mL)
0	6.37 ± 0.15	0.12 ± 0.01	0.55 ± 0.10	7.04 ± 0.95	6.98 ± 0.05
3	6.09 ± 0.13	0.14 ± 0.04	0.80 ± 0.07	6.73 ± 0.46	7.38 ± 0.40
6	5.59 ± 0.09	0.18 ± 0.01	1.12 ± 0.10	7.74 ± 0.47	7.92 ± 0.14
9	5.41 ± 0.07	0.26 ± 0.03	1.57 ± 0.07	6.76 ± 0.02	7.84 ± 0.34
12	4.68 ± 0.09	0.31 ± 0.01	1.62 ± 0.03	7.87 ± 0.34	7.89 ± 0.19
15	4.36 ± 0.17	0.37 ± 0.03	1.92 ± 0.05	8.1 ± 1.01	7.51 ± 0.04
18	4.06 ± 0.06	0.42 ± 0.01	1.89 ± 0.03	7.79 ± 0.25	7.80 ± 0.26
21	3.96 ± 0.03	0.45 ± 0.01	1.77 ± 0.06	8.00 ± 0.56	7.72 ± 0.19
24	3.9 ± 0.05	0.45 ± 0.03	2.03 ± 0.03	7.99 ± 0.40	7.78 ± 0.16
27	3.84 ± 0.06	0.49 ± 0.02	1.91 ± 0.05	8.01 ± 0.28	7.82 ± 0.08
30	3.81 ± 0.04	0.48 ± 0.02	2.01 ± 0.02	7.97 ± 0.43	7.29 ± 0.27
33	3.78 ± 0.03	0.52 ± 0.02	2.12 ± 0.09	7.92 ± 1.24	7.37 ± 0.24
36	3.77 ± 0.01	0.53 ± 0.03	2.05 ± 0.03	7.68 ± 0.60	7.81 ± 0.17

* Values are mean ± standard deviation.

**Table 2 foods-10-01447-t002:** Exponential model parameters for the kinetics of pH and TTA of millet slurry during fermentation *.

Variable	Model Parameters			R^2^
	a	b	c	
pH	3.382 ± 0.213	−3.188 ± 0.203	0.071 ± 0.013	0.971
TTA (%)	0.663 ± 0.065	0.575 ± 0.058	0.042 ± 0.009	0.981

* Values are parameter ± standard error; TTA = Titratable acidity; R^2^ = 1 − (Residual Sum of Squares)/(Corrected Sum of Squares).

**Table 3 foods-10-01447-t003:** Gompertz parameters for total viable and lactic acid bacteria population during millet slurry fermentation.

Parameter	Count (CFU/mL)	
	Lactic Acid Bacteria	Total Viable Count
K	6.971	6.911
A	0.092	0.790
µ_max_	0.092	0.202
λ (h)	3.903	0.000
E%	0.70	2.01

K = initial level of bacterial count (log CFU/mL), A = increase in log CFU/mL between time = 0 and the maximum population density at the stationary phase (log CFU/mL), µ_max_ = maximum growth rate (Δlog (CFU/mL)/h), λ = lag time (h) and t = fermentation time (h); E% = relative percent difference between experimental (O) and predicted (P) values $\sum_{1}^{n}\frac{\left|O-P\right|}{O}\times 100$

**Table 4 foods-10-01447-t004:** Tentative lactic acid bacteria isolated at different times and pH during fermentation of pearl millet slurry.

pH and lactic Acid Bacteria	Fermentation Time (h)												
	0	3	6	9	12	15	18	21	24	27	30	33	36
pH *	6.37 ± 0.15	6.09 ± 0.13	5.59 ± 0.09	5.41 ± 0.07	4.68 ± 0.09	4.36 ± 0.17	4.06 ± 0.06	3.96 ± 0.03	3.9 ± 0.05	3.84 ± 0.06	3.81 ± 0.04	3.78 ± 0.03	3.77 ± 0.01
*Leuconostoc mesenteroides* ssp. *dextranicum*	x												
*Leuconostoc pseudomesenteroides*	x	x	x										
*Pediococcus pentosaceus*	x			x			x						x
*Streptococcus thoraltensis;*	x		x						x				
*Enterococcus gallinarum*					X	x		x	x		x		
*Enterococcus casseliflavus*		x				x							
*Enterococcus faecium*					X	x	x	x	x		x		
*Enterococcus faecalis*											x		
*Enterococcus avium*				x									
*Enterococcus durans*				x									

* Values are mean ± standard deviation of triplicate readings. x—Indicates the time the bacteria was isolated.

**Table 5 foods-10-01447-t005:** Physiological properties of tentative lactic acid bacteria isolated from pearl millet slurry during fermentation.

Lactic Acid Bacteria	Gram Reaction	Catalase Test	Morphology	Hot-Loop Test	4 °C	10 °C	45 °C	6.5% NaCl
*L. mesenteroides* ssp. *dextranicum*	+	−	Cocci, groups forming chains	−	−	+	−	+
*L. pseudomesenteroides*	+	−	Cocci, groups forming chains	−	−	+	−	+
*P. pentosaceus*	+	−	Cocci, groups forming chains	−	−	+	−	+
*S. thoraltensis;*	+	−	Cocci, strepto forming chains	−	−	+	−	+
*E. gallinarum*	+	−	Cocci, groups forming chains	−	−	+	−	+
*E. casseliflavus*	+	−	Cocci, single, pairs, tetracocci forming small chains	−	−	−	−	−
*E. faecium*	+	−	Cocci, groups forming chains	−	−	+	+	+
*E. faecalis*	+	−	Cocci, groups forming chains	−	−	+	−	+
*E. avium*	+	−	Cocci, single, pairs, groups forming chains	−	−	−	−	−
*E. durans*	+	−	Cocci, groups forming chains	−	−	+	−	+

**Table 6 foods-10-01447-t006:** Effect of isolated bioburen lactic acid bacteria on the pH, titratable acidity and viscosity of fermented pearl millet extract *.

Independent Variable			Dependent Variable		
*L. mesenteroides*	*P. pentosaceus*	*E. gallinarum*	pH	Titratable Acidity (%)	Viscosity (mPa.s)
0.050	0.050	0.050	3.58 ± 0.15	0.59 ±0.06	6.68 ± 4.42
0.050	0.050	0.100	3.57 ± 0.16	0.61 ± 0.09	1.32 ± 1.76
0.050	0.100	0.050	3.63 ± 0.18	0.59 ± 0.080	6.48 ± 4.01
0.050	0.100	0.100	3.63 ± 0.20	0.56 ± 0.02	5.72 ± 5.96
0.075	0.075	0.075	3.57 ± 0.16	0.59 ± 0.04	6.71 ± 1.71
0.100	0.050	0.050	3.61 ± 0.17	0.55 ± 0.03	2.11 ± 0.84
0.100	0.050	0.100	3.76 ± 0.11	0.54 ± 0.03	5.17 ± 3.98
0.100	0.100	0.050	3.52 ± 0.14	0.61 ± 0.02	4.62 ± 4.84
0.100	0.100	0.100	3.64 ± 0.20	0.59 ± 0.06	10.67 ± 0.81

* Values are mean ± standard deviation of triplicate readings.

**Table 7 foods-10-01447-t007:** The generalized linear model for the effects of *L. mesenteroides P. pentosaceus* and *E. gallinarum* and their interaction on the pH of pearl millet extract.

Parameter	Coefficient (*β*)	Std, Error	95% Wald Confidence Interval		Significance
			Lower	Upper	
Linear coefficient effect					
Intercept	3.44	0.06	3.32	3.57	0.000
Main coefficient effect					
*L. mesentoroides* (X_1_)	1.28	0.53	0.24	2.31	0.016
*P. pentosaceus* (X_2_)	4.41	0.53	3.37	5.45	0.000
*E. gallinarum* (X_3_)	−2.73	0.53	−3.76	−1.69	0.000
Interactive coefficient effect					
*L. mesenteroides × P. pentosaceus*	−63.00	4.85	−72.51	−53.49	0.000
*L. mesenteroides* × *E. gallinarum*	55.00	4.85	45.49	64.51	0.000
*P. pentosaceus* × *E. gallinarum*	−2.33	4.85	−11.85	7.18	0.631
(Scale)	0.027				

**Table 8 foods-10-01447-t008:** The generalized linear model for the effects of *L. mesenteroides, P. pentosaceus* and *E. gallinarum* and their interaction on the total titratable acidity (TTA) of pearl millet extract.

Parameter	Coefficient (β)	Std, Error	95% Wald Confidence Interval		Significance
			Lower	Upper	
Linear coefficient effect					
(Intercept)	0.686	0.035	0.616	0.755	0.000
Main coefficient effect					
*L. mesentoroides* (X_1_)	−2.383	0.311	−2.993	−1.774	0.000
*P. pentosaceus* (X_2_)	−1.15	0.311	−1.759	−0.541	0.000
*E. gallinarum* (X_3_)	1.05	0.311	0.441	1.659	0.000
Interactive coefficient effect					
*L. mesenteroides* × *P. pentosaceus*	32	2.853	26.408	37.59	0.000
*L. mesenteroides* × *E. gallinarum*	−4.667	2.853	−10.26	0.925	0.102
*P. pentosaceus* × *E. gallinarum*	−12.667	2.853	−18.26	−7.075	0.000
(Scale)	0.003				

**Table 9 foods-10-01447-t009:** The generalized linear model for the effects of *L. mesenteroides, P. pentosaceus* and *E. gallinarum* and their interaction on the viscosity of pearl millet extract.

Parameter	Coefficient (*β*)	Std. Error	95% Wald Confidence Interval		Significance
			Lower	Upper	
Linear coefficient effect					
(Intercept)	34.44	2.41	29.71	39.16	0.000
Main coefficient effect					
*L. mesenteroides* (X_1_)	−347.18	30.41	−406.79	−287.58	0.000
*P. pentosaceus* (X_2_)	−183.45	30.41	−243.05	−123.85	0.000
*E. gallinarum* (X_3_)	−400.45	30.41	−460.05	−340.85	0.000
Interactive coefficient effect					
*L. mesenteroides and P. pentosaceus*	1742.13	384.66	988.22		0.000
*L. mesenteroides and E. gallinarum*	4020.87	384.66	3266.95	4774.78	0.000
*P. pentosaceus and E. gallinarum*	2494.20	384.66	1740.29	3248.11	0.000
*L. mesenteroides,* *P. pentosaceus and E. gallinarum*	−13,017.33	4865.56	−22,553.67	−3481.00	0.007
(Scale)	12.49				

## Data Availability

No new data were created or analysed in this study. Data sharing does not apply to this article.

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
