# Peer review of "Non-Alcoholic Pearl Millet Beverage Innovation with Own Bioburden: Leuconostoc mesenteroides, Pediococcus pentosaceus and Enterococcus gallinarum"

_foods, 2021, doi:10.3390/foods10071447_

Round 1

Reviewer 1 Report

The manuscript titled " Non-alcoholic Pearl Millet Beverage Innovation With Own Bioburden: Leuconostoc mesenteroides, Pediococcus pentosaceus and Enterococcus gallinarum” contains informations on the characterization of LAB involved in the natural fermentation of pearl millet slurry and use of starter culture for the production of a plant-based beverage.

A lot of experiments have been realized and i have some major concerns about the methodology.

Part 2.4 : The growth of lactic acid bacteria was determined in triplicates using optical density. However, a lot of genera, species and strains are present in PMS. It is not relevant to measure OD of a mixed culture. Thus, it is not possible, in my opinion, to discuss results.

Moreover,  i don’t understand how OD was estimated. Line 106, it is written that samples were drawn during the fermentation of PMS for OD. But, line 129, it is written that 1 mL of the slurry was transferred into MRS broth and incubated at 30°C for 48h. I understood that OD was estimated during the growth of LAB in MRS ? Is that exact ? It is not clear.

If OD was determinated during the incubation of MRS broth, why did you choose to study the growth of LAB at 30°C (line 130). Fermentation of PMS was realized at 37°C.

Why did you discuus results of OD, pH and TTA while OD values were estimated in MRS broth and pH&TTA values in PMS ?

And, i don’t understand why it is important to measure OD. Enumeration of LAB on MRS agar is also realized. It gives informations about growth

Part 2.6 Isolation and identification of lactic acid bacteria in pearl millet slurry during fermentation. Line 169 : The colonies were identified using Vitek 2 compact system. What is the principle of this system? No information is given and this system is not well known. It is difficult to judge the results without knowing the method.

I don't understand the interest of data modeling (Gompertz,..). What does modeling bring to the study? 

Table 2 : How did you calculate the µmax ? You write « μmax = maximum growth rate (Dlog (CFU/ml)/day) ». µmax was calculated / day and not/min ? Exponential growth phase lasts 24h ? It is not clear for me.

Moreover, if I understand correctly, you caculated a µmax value of a mixed culture of LAB ? µmax is strain specific and it is not relevant for a mix of strains. Thus, it is not possible to discuss results.

I have a question about reproductability of data. I don’t understand if one PMS or 3 PMS (triplicates) was/were used. It is written line 239 that “The results were reported as mean ± standard deviation of three independent trials.” Is is biological triplicate or analytical triplicate?Please indicate this point. Is is 3 PMS but not 3 experiments for the effect of bioburden LAB on pearl millet (you used a 3 level augmented factorial design)? It is not clear for me.

Specific comments :

Line 71 : All the LAB are not probiotic but some of them could be considered as probiotics

Line 129 : Are you sure that other bacteria than LAB are not able to grow in MRS broth ? Are you sure than LAB (such as Lactococci) are able to grow in MRS ? Why M17 was not added to the study ?

Line 150 : Why did you choose to incubate PCA at 37°C and not 30°C ?

Line 167 : why did you study the effect of NaCl content on growth ?

It is not easy to analyze results. As example, data about the growth of LAB are lines 263-265 and lines 347-352. Maybe, it would be useful to have a table with times, OD, log, pH, TTA, glucose.

The texte line 347 to 361 can be replaced by a table.

Part 3.4 : Does the identification provide information on the quantity os pecies (dominant or not)?

Line 443 : Pediococcus pentosaceus shown in Figure 5b was isolated at 0, 9, 18 and 36 h. How do you explain that P. pentosaceus was not isolated at 3, 6, 12, 15 and between 21-33h ? Do you think that it is linked to the method of isolation and determination ?

Figure 5 : I did not fully understand the contribution of scanning electron microscopy.

Line 97 : You write that these bacteria are not able to grow at 45°C. Are they mesophilic ou thermophilic LAB ? It is known that Leuconostoc is not able to grow at 45°C. Why did you choose to study the impact of temperature ?

Author Response

Thank you very much for your useful comments.  Please see the attachment for our responses.  We hope you will find them satisfactory.

Reviewer 2 Report

Jideani et al. described a pearl miller fermented beverage carried out with the use of own starter cultures. The paper is well written and results well presented. However, I missed in some step the reason why some of the analysis were performed and also the final aim of the study and the future prospectives.

Please find below my comments: 

  • In the introduction and in the abstract, a description and an introduction of the mathematical models applies in the study is missing. By reading it, one would think that the study is a poor microbiological study while it is not since a lot of modeling and assumptions are taken. I suggest to rework abstract and intro to make things clearer 
  • For some of the measured parameters in the study, it is not clear why the were measured. Was that to fill in the model? Please clarify the general methodological approach before moving to the small steps 
  • Why was the OD measured instead of selective plate counting for LAB strains? Was the cell concentration data needed for the model? Please elaborate on that to make it clearer 
  • What was the final scope of the observation of colony under scanning electron microscope? 
  • Figure 2 and 3 and 4: are those data only from 1 fermentation? 
  • Figure 4. The explanation given for the flat viable cell count compared to the LAB counts is not 100 % convincing me. LAB are around 8 log, while TVC are around 7.6 Isn't something missing here? 
  • Table 4. As mentioned by the authors, E. faecium is a AMR carrying species, would have been interesting to sequence the isolates and investigate the AMR profile 
  • Table 4 . It is a bit surprisingy to see species that appear and disappear (ex. pediococcus pentosaceus)... please elaborate more on that. What was the detection limit of your system? 
  • Conclusions are not exhaustive, they represent a summary of the study but there is not a real conclusion
  • Some future perspectives will be needed to give more visibility and importance to the study.  

Author Response

Thank you very much for the valuable comments.  Please see the attachment for our responses.  We hope you will find them satisfactory.

Round 2

Reviewer 1 Report

I have checked the revised version of the manuscript and  the report of authors. It is very clear now.

I have no more comments or questions.

A comment: 2.4 is 2.3; 2.5 is 2.4....
